# Theoretical Prediction on the New Types of Noble Gas Containing Anions OBONgO^−^ and OCNNgO^−^ (Ng=He, Ar, Kr and Xe)

**DOI:** 10.3390/molecules25245839

**Published:** 2020-12-10

**Authors:** Cheng-Cheng Tsai, Yu-Wei Lu, Wei-Ping Hu

**Affiliations:** Department of Chemistry and Biochemistry, National Chung Cheng University, Chia-Yi 621, Taiwan; tsai.chengcheng@gmail.com (C.-C.T.); jacky850822@gmail.com (Y.-W.L.)

**Keywords:** noble gas anions, noble gas chemistry, electron density, stability of noble gas containing molecules

## Abstract

The fluorine-less noble gas containing anions OBONgO^−^ and OCNNgO^−^ have been studied by correlated electronic structure calculation and density functional theory. The obtained energetics indicates that for Ng=Kr and Xe, these anions should be kinetically stable at low temperature. The molecular structures and electron density distribution suggests that these anions are stabilized by ion-induced dipole interactions with charges concentrated on the electronegative OBO and OCN groups. The current study shows that in addition to the fluoride ion, polyatomic groups with strong electronic affinities can also form stable noble gas containing anions of the type Y^−^…NgO.

## 1. Introduction

Noble gas (Ng) atoms contain completely filled valence shells and thus do not participate in the normal chemical bonding that the other main group atoms do. Studies since the 1960s, however, have shown that noble gases do form chemical compounds with very electronegative atoms and groups, such as in XePtF_6_ [1], XeF_2_ [2], XeF_4_ [3], XeO_4_ [4], etc. From the 1980s, chemists have successfully made various noble gas hydrides HNgX in noble gas matrixes at cryogenic condition [5,6,7,8,9], including the only known neutral argon-containing molecules HArF [6]. In recent years, new experimental techniques and computational studies have discovered and predicted many new types of noble gas containing molecules, and the field of noble gas chemistry has been flourishing [10,11,12,13,14,15].

Since the inertness of noble gases derives from the filled valence electrons, it is not surprising that varieties of noble gas containing cations can be made, such as [UO_2_ (Ng)*_n_*]^+^ [16], HNgFNgH^+^ [17], HNgCS^+^ [18], and HNgCCO^+^ [19]. On the other hand, the formation of noble gas containing anions are expected to be more difficult since there are “too many” electrons in noble gases for chemical bonding already. However, several types of stable anions have been predicted such as FNgO^−^ [20], FNgBN^−^ [21], FNgCC^−^ [22], NNgO_3_^−^ [23], [NgBeB_11_(CN)_11_]^2−^ [24], [B_12_Ng_12_F_12_]^2−^ [25], XAuNgX^−^ [26], FNgS^−^ [27], [B_12_C_11_Ng]^−^ [28], and [B_12_(CN)_11_Ng]^−^ [29,30]. Through some inductive effects, the binding of the excess electrons and the chemical bonding is surprisingly strong. It thus seems that there is rich anionic noble gas chemistry after all.

For the FNgX^−^ (X=O, BN, CC) anions, the stability comes from the ion induced dipole interaction. That is, the high charge density of the fluoride ion strengthens the NgX bonding with significant electron density transfer from the noble gas atom to the X group, while the fluorine atom remains its anionic character. The three-body dissociation of FNgX may proceed through two reactions:FNgX^−^ → F^−^ + Ng + X → Erxn = ΔE1(R1)
FNgX^−^ → F + Ng + X^−^ → Erxn = ΔE2(R2)
and the difference in the energies of reaction (Erxn) is just the difference of the electron affinity (EA) between F and X:F^−^ + X → F + X^−^ + Erxn = ΔE2 − ΔE1 = EA(F) − EA(X)(R3)
where EA(X) is defined by
X^−^ → X + e^−^ → Erxn = EA(X)(R4)

For example, from a previous computational study on FHeO^−^ [20], ΔE1 = 20.5 kcal/mol, EA(F) − EA (O singlet) = −4.7 kcal/mol, and thus ΔE2 = 15.9 kcal/mol. The electron affinity of a singlet oxygen atom is higher than that of the fluorine atom, and thus the stability against the three-body dissociation is determined by (R2). It is interesting to know whether other atomic or polyatomic anions Y^−^ can replace the fluoride and also form stable anion YNgX^−^. From above, the Y^−^ must have strong polarizing effects with a charge concentrated on a small atom (O, N) to give a relatively large ΔE1. In addition, Y must have an EA larger or comparable to that of X, otherwise the (R2) could become a fast dissociation channel due to the low energy of reaction (ΔE2). For stability, the exoergic two-body dissociation channel YNgX^−^ → Ng + YX^−^ also needs to be considered [20,21,22]. If the barrier height is low, the anion would dissociate quickly even though the three-body dissociation energies (ΔE1, ΔE2) are high. Additionally, the YNgX^−^ may be susceptible to intersystem crossing to a repulsive triplet state at larger Ng–X distances [20].

The OCN and OBO are isoelectronic groups with very high electron affinities. In the current study, we investigated whether these groups can form stable noble gas containing anions (OBO^−^…NgO, OCN^−^…NgO, NCO^−^…NgO) and analyzed their molecular structures and chemical bonding.

## 2. Methods

Molecular geometry and harmonic vibrational frequencies were calculated using the MP2 theory [31] with the aug-cc-pVDZ and aug-cc-pVTZ basis sets [32,33]. They were also obtained using the B3LYP [34] and MPW1B95 [35] hybrid density functional theory with the aug-cc-pVTZ basis set. A recent benchmark study [36] has shown that the MPW1B95 functional gave a very accurate prediction of the noble gas bonding energies. Single-point energy calculation was also performed using the CCSD(T) theory [37] with the aug-cc-pVTZ and aug-cc-pVQZ basis sets to acquire more accurate relative energies. For the Ar atom, the related basis sets aug-cc-pV(D + d)Z, aug-cc-pV(T + d)Z, and aug-cc-pV(Q + d)Z [38] were used; and for the Xe atom, the basis sets aug-cc-pVDZ-pp, aug-cc-pVTZ-pp, and aug-cc-pVQZ-pp [39] were used where the inner 28 electrons were replaced by a relativistic pseudopotential. In the rest of this article, the basis sets will be abbreviated as ap*n*z (*n* = d, t, q). Since in the following discussion the relative energies will be based on the CCSD(T) calculation with a very large basis set (apqz), and there are no complexation energies involved in (R1) and (R2), the basis-set superposition errors are not expected to be significant. The electron density distribution and ChelpG atomic charges [40] were obtained at the MP2/aug-cc-pVDZ level. All the electronic structure calculation was performed using Gaussian 09 program rev. D01 [41].

## 3. Results and Discussion

### 3.1. Structures

Calculated structures of OBONgO^−^ (Ng=He, Ar, Kr, Xe) at the MP2/aptz level are shown in Figure 1. Structures obtained at other theoretical levels are included in the Appendix A. Interestingly, for Ng=He and Xe the predicted structures are bent, but for Ar and Kr the calculated structures are linear. This is due to the very small force constants along the Ng–O–B angle where the Ng–OBO bonding is mainly ionic. At the highest level of theory in the current study, the energy differences between linear and bent structures are within 1 kcal/mol. This will be discussed later in this article. Structures predicted by the B3LYP and MPW1B95 density functional theory are all nonlinear. In all calculated bent conformation, the Ng–O–B angles were predicted from 120 to 135 degrees. The terminal Ng–O distances were predicted to be 1.064–1.899 Å, which are quite short and correlate well with the sizes of the noble gas atoms. They are similar to the Ng-distances in the FNgO^−^ anions from a previous study [20]. The Ng–OBO distance was predicted to be from 1.852 to 2.509 Å. It is noted that this distance increases only slightly (0.05 Å) from OBOArO^−^ to OBOXeO^−^, which is consistent with the bonding being ionic in nature. Thus, the structures are better represented as [OBO…Ng=O]^−^ In all cases, the O–B distance on the noble gas side is slightly (0.01–0.02 Å) longer than the distance on the other side. Both distances are very close to the O–B distance (1.275 Å) in OBO^−^.

Calculated structures of OCNNgO^−^ (Ng=He, Ar, Kr, Xe) at the MP2/aptz level are shown in Figure 2. Structures obtained at other theoretical levels are also included in the Appendix A. All predicted structures are planar and bent except for OCNHeO^−^ which is linear. Structures predicted by density functional theory are all nonlinear. The predicted terminal Ng–O distances are very similar to those in OBONgO^−^ The N–Ng distances were predicted slightly longer (0.02–0.06 Å) than the corresponding O–Ng distances in OBONgO^−^. This distance is also insensitive to the identity of the noble gas, and it increases only 0.02 Å from Ng=Ar to Ng=Xe. The calculated C–N distances are slightly shorter (0.01–0.02 Å) than the O–C distances. Both distances are very similar to the corresponding bond distance (O–C = 1.233 Å, C–N = 1.204 Å) in OCN^−^.

The structures of the isomeric NCONgO^−^ are shown in Figure 3. All predicted structures are planar and bent. In NCONgO^–^, the predicted differences in the C–N and C–O distances are more significant with the C–O distances, being only 0.04–0.05 Å longer. As compared to OCNNgO^−^, the O–C distances are 0.02–0.03 Å longer, the C–N distances ~0.01 Å shorter, and the terminal Ng–O distances are very similar.

The transition state (TS) structures for the two-body dissociation reactions OBONgO^−^ → Ng + BO_3_^−^ are shown in Figure 4. The significant elongation of the OBO…Ng distances relative to those in OBONgO^−^ are observed. For Ng=He, the NgO distance in TS also increases significantly. The Ng–O–B bond angles in the TS decrease to ~80 degrees, except for Ng=He. Transition state structures for the two-body dissociation of OCNNgO^−^ are included in the Appendix A.

### 3.2. Energetics

The predicted energies of reactions and dissociation barrier heights are listed in Table 1. The calculated electron affinities of the O, OBO, and OCN at CCSD(T)/apqz level are 82.6, 105.8, and 83.8 kcal/mol, respectively. According to (R1–R4), the stability against the three-body dissociation of OBONgO^−^ and OCNNgO^−^ is determined by the reactions:OBONgO^−^ → OBO^−^ + Ng + O(R5)
OCNNgO^−^ → OCN^−^ + Ng + O(R6)

As shown in Table 1, the energy of reactions at the CCSD(T)/apqz level for (R5) are 7.4, 26.5, 43.6, and 67.4 kcal/mol for Ng=He, Ar, Kr, Xe, respectively. The zero-point energies make a significant correction for Ng=He on the three-body dissociation channel. The values obtained at the MP2/aptz level overestimate the stability by 13–21 kcal/mol, which is consistent with earlier studies [20,22,36,42]. The barrier heights of the two-body dissociation at the CCSD(T)/apqz level are 6–32 kcal/mol. The calculation thus indicates that for Ng=Ar, Kr, and Xe, the OBONgO^−^ are kinetically stable at low temperature, while for Ng=He, the anion is only marginally stable at the lowest temperature.

The OCNNgO^−^ and NCONgO^−^ are predicted to be very close in energies. All calculations predict that OCNNgO^−^ are slightly lower in energies (data included in Appendix A), and at the CCSD(T)/apqz level, the differences are only 2–4 kcal/mol. The interconversion reaction barriers from NCONgO^−^ to OCNNgO^−^ were estimated to be only 1–4 kcal/mol. The interconversion TS structures and the reaction energetics are included in the Appendix A. The predicted energies and barrier heights for the unimolecular dissociation reactions of OCNNgO^−^ are listed in Table 2. Predicted energetics for NCONgO^−^ is included in Appendix A. As shown in the Table 2, the energy of reactions at CCSD(T)/apqz level for (R6) are 7.2, 26.5, 43.9, and 67.8 kcal/mol for Ng=He, Ar, Kr, and Xe, respectively. They are very similar to those for OBONgO^−^. The barrier heights of the two-body dissociation at CCSD(T)/aptz level are also very similar to those for OBONgO^−^. Thus, the OBONgO^−^, OCNNgO^−^, and NCONgO^−^ all have very similar kinetic stability. As compared to FNgO^−^ these anions show higher stability against the three-body dissociation but slightly lower two-body dissociation barriers. As in the cases for FArO^−^, molecules with covalent ArO bonding are susceptible to intersystem crossing to the repulsive triplet state [20]. Figure 5 shows the calculated singlet and triplet potential energy curves along the terminal Ar–O bond of OBOArO^−^, and the crossing point is only approximately 6 kcal/mol above the singlet minimum, similar to that in FArO^−^. For Ng=Kr and Xe, the crossing points are located 15–23 kcal/mol above the singlet minima (data not shown), and the singlet–triplet gaps calculated at the singlet minima are 51–55 kcal/mol, which are ~10 kcal/mol higher than that of Ar. Thus, for Ng=Kr and Xe, these anions are less susceptible to the dissociation through intersystem crossing.

### 3.3. Electron Density

The ChelpG atomic charges for OBONgO^−^ are shown in Figure 1. The negative charges are clearly concentrated on the OBO groups. Plots of electron density distribution and Laplace concentration [43,44,45] are shown in Figure 6 and Figure 7, respectively. The lack of electron density between the OBO and NgO groups suggests mainly ionic interaction. Since ionic interaction is non-directional, the force constants along the Ng–O–B angles are typically small, and different theoretical methods would likely predict different optimized angles (120–180 degrees). The significant distortion in electron density along the bonding direction indicates that the Ng–O and O–B–O bonds are polar covalent [20,21,22,46,47]. The increasing polarity from He–O to Xe–O is also evident. Similar information is obtained from Figure 7 where electron concentration is seen between B–O (and, to a lesser extent, Ng–O), and electron depletion is observed between OBO and NgO. The depletion is less evident for the larger noble gases. While a more sophisticated analysis on the chemical bonding of the noble gas containing molecules has been recently reported [48], the current study focuses more on the quantum mechanical “observables” such as stability (energies), molecular structures, and electron densities. More detailed bonding analysis on these anions will be performed in future studies.

From the calculated atomic charges for OCNNgO^−^ in Figure 2, the negative charges are also concentrated on the OCN groups. Plots of electron density distribution and Laplace concentration are shown in Figure 8 and Figure 9, respectively. The lack of electron density between the OCN and NgO groups again suggests that the interaction is ionic. Figure 8 shows that the polarity of O–C is much stronger than C–N, as expected. Figure 9 shows significant charge concentration within the OCN groups, and the concentration region around C–N is slightly larger than that around C–O. The atomic charges and electron density plots of NCONgO^−^ are included in the Appendix A.

## 4. Conclusions

The anions OBONgO^−^, OCNNgO^−^, and NCONgO^−^ (Ng=He, Ar, Kr, Xe) have been studied by correlated electronic structure calculation. The results show that they are similar to FNgO^−^ with very short terminal NgO distances, and with the negative charge concentrated on the OBO (OCN, NCO) groups. The structures can conceptually be written as Y^−^…Ng=O where the interaction between Y and NgO is mostly ionic and the Ng=O bonding is polar covalent. The anions are predicted to be kinetically stable for Ng=Ar, Kr, and Xe on the singlet potential energy surface with the Ar-containing anions susceptible to intersystem crossing. The current study suggests that electronegative, fluorine-less chemical groups can form stable noble gas containing anions through ion-induced polarization, and these Y^−^…Ng=O anions could be detectable in future experiments under cryogenic conditions.

## Figures and Tables

**Figure 1 molecules-25-05839-f001:**
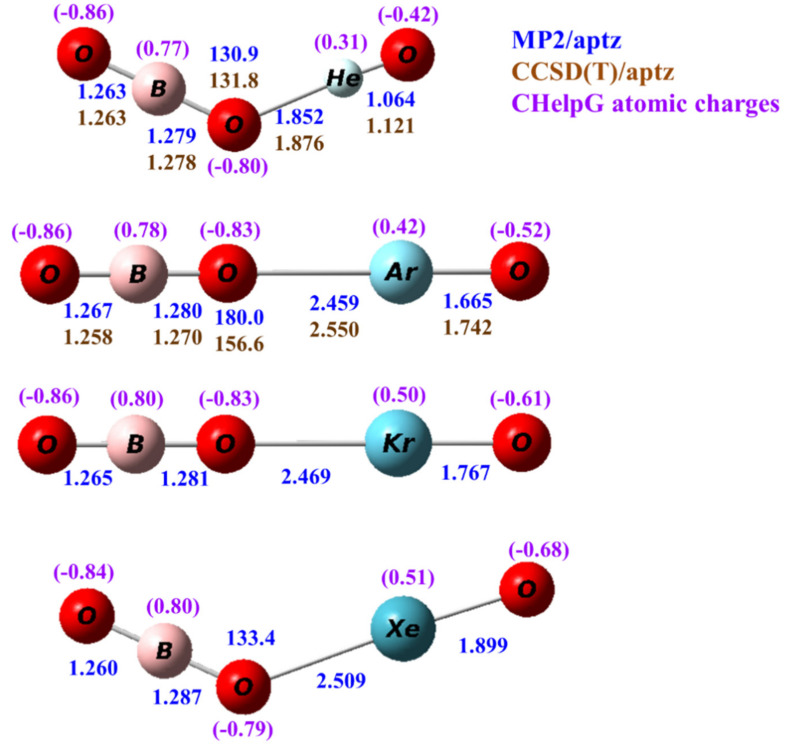
Calculated structures of OBONgO^−^ (Ng=He, Ar, Kr and Xe). The bond distances are in angstroms and the bond angles in degrees. The numbers in blue and brown are values calculated by the MP2/aptz and CCSD(T)/aptz methods, respectively. The values in purple are CHelpG atomic charges.

**Figure 2 molecules-25-05839-f002:**
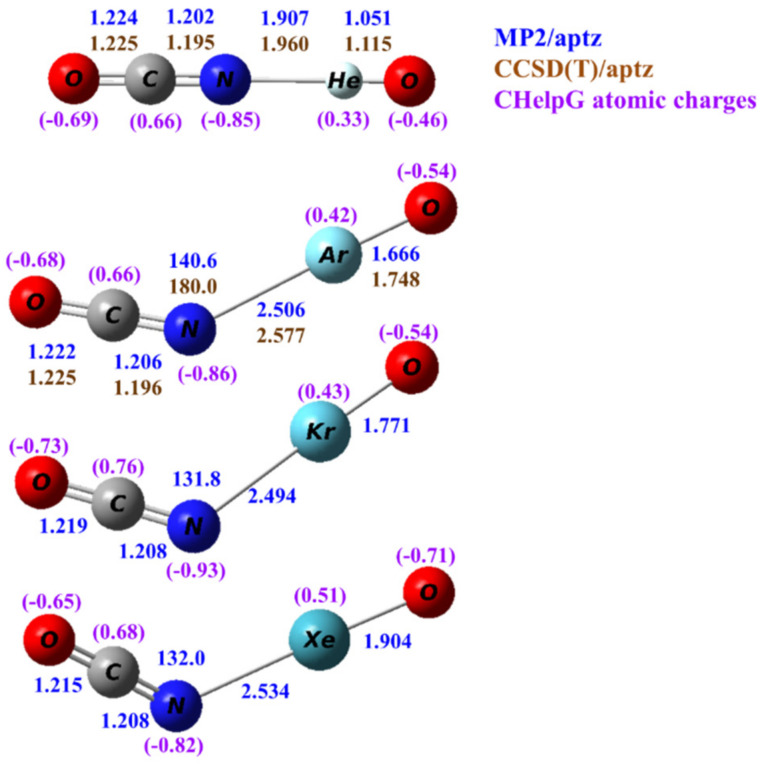
Calculated structures of OCNNgO^−^ (Ng=He, Ar, Kr and Xe). The bond distances are in angstroms and the bond angles in degrees. The numbers in blue and brown are values calculated by the MP2/aptz and CCSD(T)/aptz methods, respectively. The values in purple are CHelpG atomic charges.

**Figure 3 molecules-25-05839-f003:**
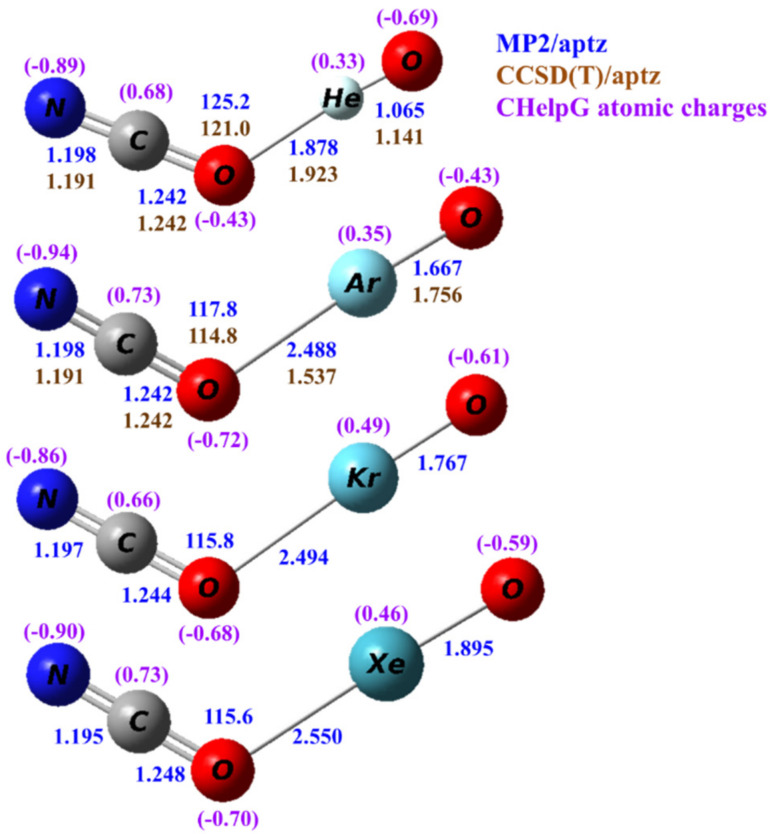
Calculated structures of NCONgO^−^ (Ng=He, Ar, Kr and Xe). The bond distances are in angstroms and the bond angles in degrees. The numbers in blue and brown are values calculated by the MP2/aptz and CCSD(T)/aptz methods, respectively. The values in purple are CHelpG atomic charges.

**Figure 4 molecules-25-05839-f004:**
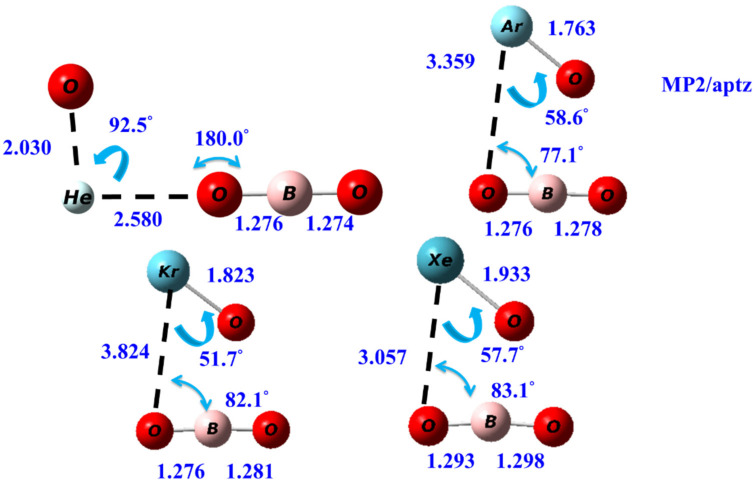
The calculated transition state structures of the two-body dissociation reactions of the OBONgO^−^ at MP2/aptz level.

**Figure 5 molecules-25-05839-f005:**
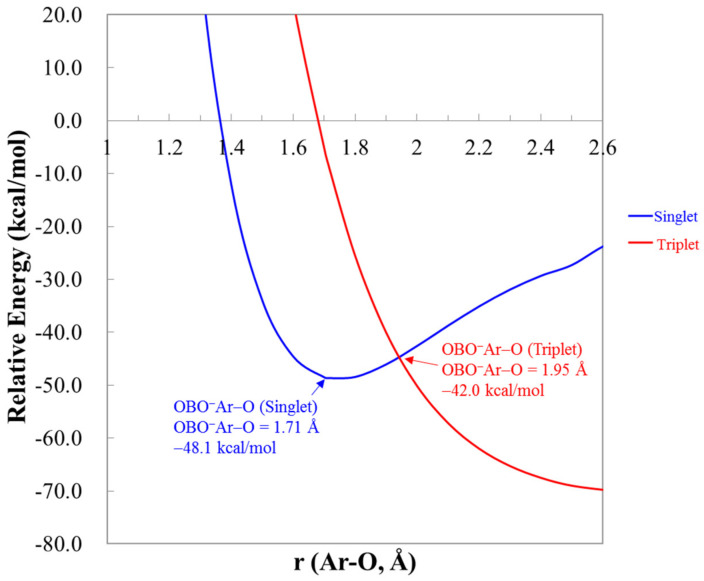
The singlet and triplet potential energy curves of OBOArO^−^ by extending the terminal Ar–O bond at the CCSD(T)/aptz level. The zero of the energy is the total energy of a singlet oxygen atom, an OBO anion, and an argon atom.

**Figure 6 molecules-25-05839-f006:**
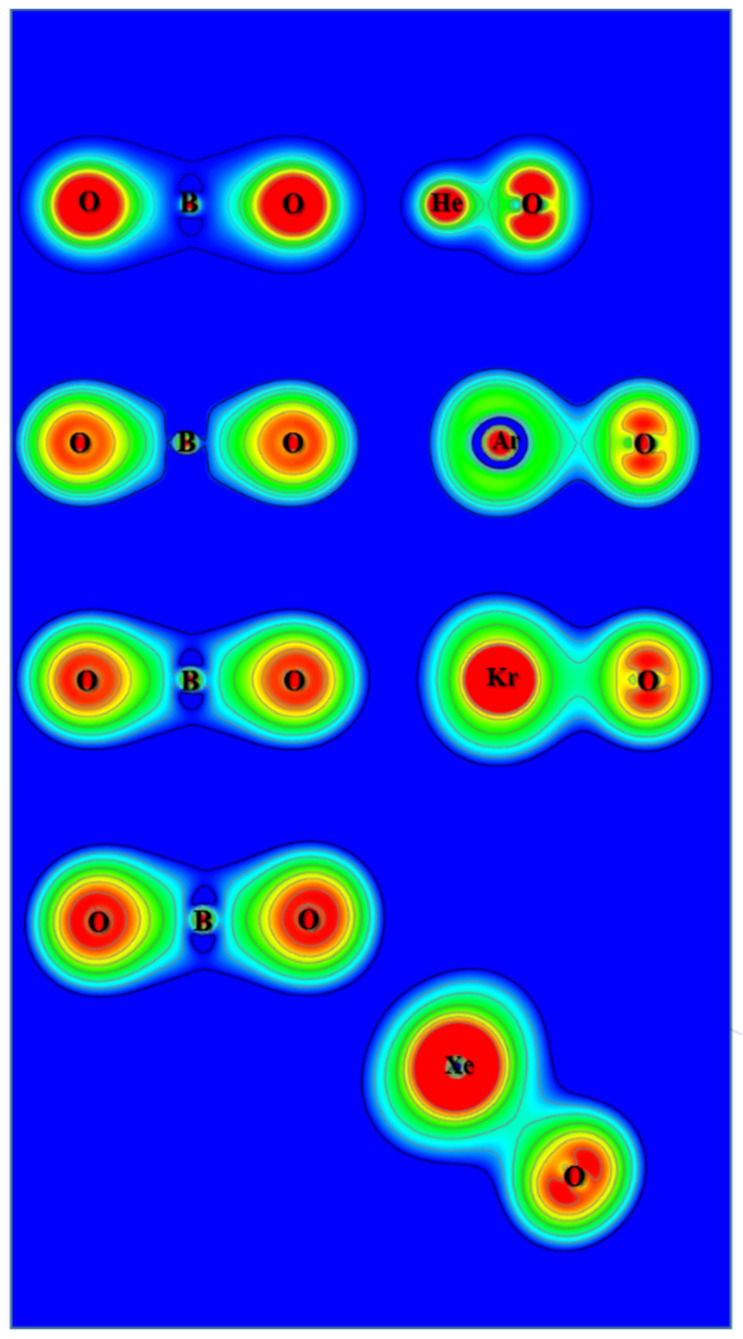
Contour plots of the calculated electron density of OBONgO^−^.

**Figure 7 molecules-25-05839-f007:**
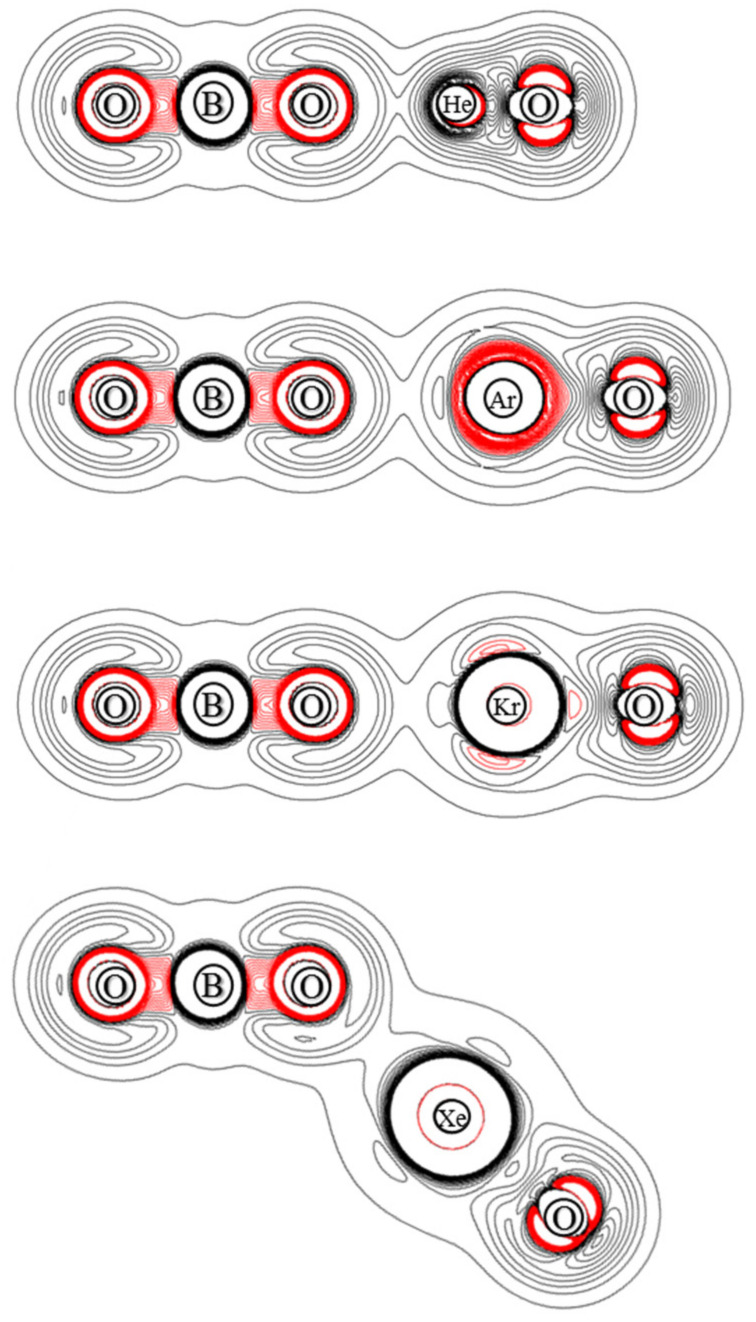
Contour plots of the calculated Laplace concentration of OBONgO^−^. The red contour lines are in regions of charge concentration and the black contour lines are in regions of charge depletion.

**Figure 8 molecules-25-05839-f008:**
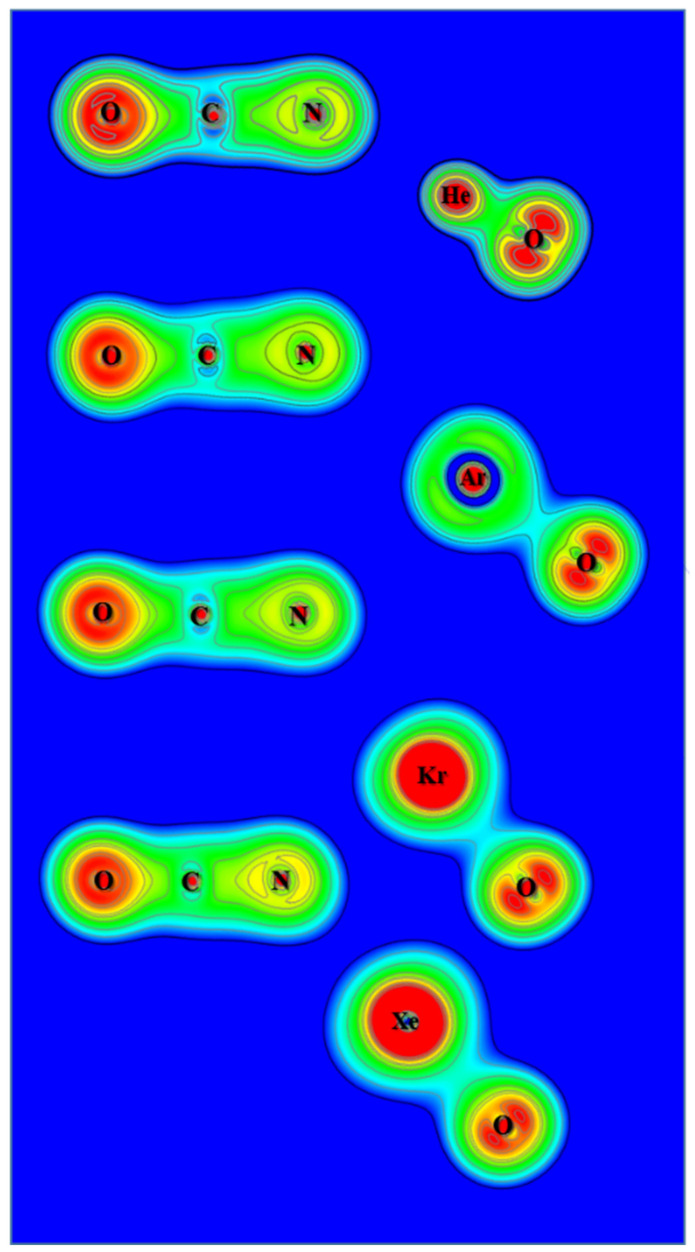
Contour plots of the calculated electron density of OCNNgO^−^.

**Figure 9 molecules-25-05839-f009:**
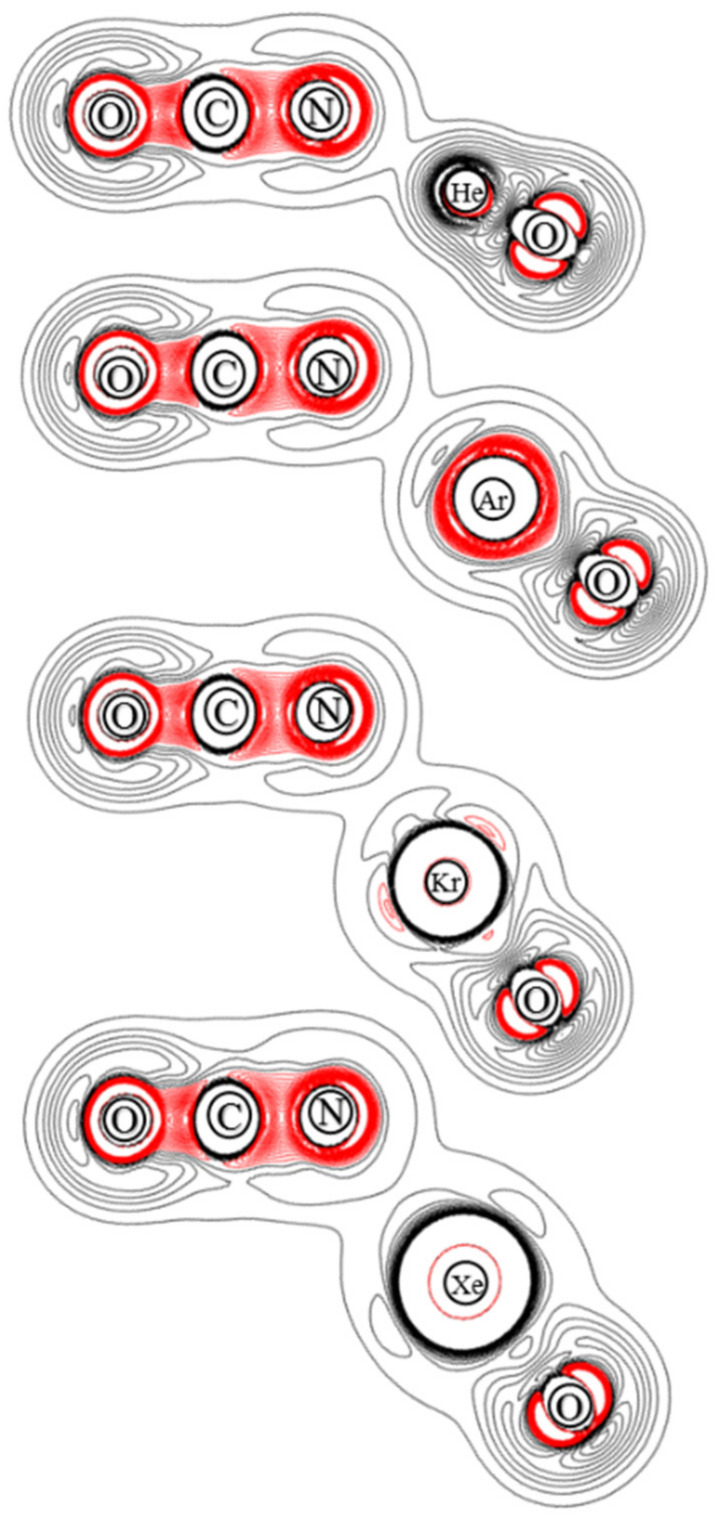
Contour plots of the calculated Laplace concentration of OCNNgO^−^. The red contour lines are in regions of charge concentration and the black contour lines are in regions of charge depletion.

**Table 1 molecules-25-05839-t001:** The calculated three- and two-body dissociation energies, the two-body dissociation barriers, and the vertical singlet-triplet gaps of OBONgO^−^. Energies listed are Born-Oppenheimer energies in kcal/mol, and the values in parentheses include zero-point energy correction.

OBONgO^−^	OBO^−^ + Ng + O	Ng + BO_3_^−^	Barrier	S–T Gap
Ng=He				
MP2/apdz	15.4 (11.8)	−96.7 (−95.9)	13.6 (10.4)	82.5
MP2/aptz	19.8 (16.1)	−94.7 (−94.0)	17.6 (14.3)	95.7
CCSD(T)/aptz	6.4 (3.8)	−72.6 (−71.8)	N.A. ^b^	66.3
CCSD(T)/aptz ^a^	6.1	−72.1	5.2	91.7
CCSD(T)/apqz ^a^	7.4	−70.3	6.4	80.2
Ng=Ar				
MP2/apdz	32.6 (31.7)	−79.5 (−76.0)	21.2 (20.4)	39.4
MP2/aptz	41.3 (39.5)	−73.2 (−70.6)	23.6 (22.8)	42.6
CCSD(T)/aptz	26.7 (25.0)	−74.8 (−73.5)	N.A. ^b^	30.4
CCSD(T)/aptz ^a^	25.5	−52.6	18.1	41.1
CCSD(T)/apqz ^a^	26.5	−51.2	18.5	41.3
Ng=Kr				
MP2/apdz	51.6 (50.0)	−60.4 (−57.8)	28.0 (27.4)	50.5
MP2/aptz	60.7 (59.0)	−53.7 (−51.0)	30.0 (29.3)	62.4
CCSD(T)/aptz ^a^	42.5	−35.7	24.3	58.9
CCSD(T)/apqz ^a^	43.6	−34.1	24.7	49.8
Ng=Xe				
MP2/apdz	76.2 (74.4)	−35.9 (−33.3)	35.7 (34.9)	67.1
MP2/aptz	86.8 (85.5)	−27.6 (−24.6)	37.6 (37.3)	69.2
CCSD(T)/aptz ^a^	65.8	−12.3	31.9	55.5
CCSD(T)/apqz ^a^	67.4	−10.3	32.4	55.4

^a^ single-point calculation using MP2/apdz structures. ^b^ not available.

**Table 2 molecules-25-05839-t002:** The calculated three- and two-body dissociation energies, the two-body dissociation barriers, and the vertical singlet-triplet gaps of OCNNgO^−^. Energies listed are Born-Oppenheimer energies in kcal/mol, and the values in parentheses include zero-point energy correction.

OCNNgO^−^	OCN^−^ + Ng + O	Ng + NC(OO)^−^	Barrier	S–T Gap
Ng=He				
MP2/apdz	15.8 (12.1)	−55.7 (−57.9)	14.0 (11.1)	85.9
MP2/aptz	17.4 (16.5)	−58.2 (−57.6)	17.5 (14.5)	88.0
CCSD(T)/aptz	7.0 (3.9)	−54.1 (−55.4)	N.A. ^b^	69.3
CCSD(T)/aptz ^a^	6.6	−53.8	5.6	115.6
CCSD(T)/apqz ^a^	7.2	−53.1	11.4	89.4
Ng=Ar				
MP2/apdz	33.3 (32.1)	−38.3 (−37.9)	22.1 (21.2)	41.1
MP2/aptz	42.3 (40.5)	−33.3 (−33.7)	24.9 (24.0)	55.8
CCSD(T)/aptz	27.3 (25.0)	−36.9 (−34.3)	N.A. ^b^	34.4
CCSD(T)/aptz ^a^	26.2	−34.2	19.1	42.6
CCSD(T)/apqz ^a^	26.5	−33.9	19.5	42.8
Ng=Kr				
MP2/apdz	53.1 (51.0)	−18.5 (−19.0)	30.0 (29.1)	52.5
MP2/aptz	62.6 (60.7)	−13.3 (−13.4)	32.1 (31.3)	64.5
CCSD(T)/aptz ^a^	43.6	−16.8	26.3	51.3
CCSD(T)/apqz ^a^	43.9	−16.4	26.6	51.4
Ng=Xe				
MP2/apdz	78.3 (76.2)	6.8 (6.2)	38.7 (37.9)	61.6
MP2/aptz	89.7 (87.7)	14.1 (13.6)	41.1 (40.2)	71.4
CCSD(T)/aptz ^a^	67.0	6.4	34.1	56.8
CCSD(T)/apqz ^a^	67.8	7.4	32.4	55.4

^a^ single-point calculation using MP2/apdz structures. ^b^ not available.

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
