# Peer review of "Theoretical Prediction on the New Types of Noble Gas Containing Anions OBONgO and OCNNgO (Ng=He, Ar, Kr and Xe)"

_molecules, 2020, doi:10.3390/molecules25245839_

Round 1

Reviewer 1 Report

In this paper the authors reported wave function based calculations on the structure and energetics of several noble-gas containing anions and discussed about their stability. The study closely resembles the work on the FNgO- anions of the same research group published in 2005 in both the methods and the results. While the polyatomic groups clearly make the calculations more demanding, they still follows standard procedures and the authors did not make use of the more advanced tools offered by recent developments in quantum chemistry.

That said, the results support the conclusion that the anions are kinetically stable and that the bonding can be considered as the charge--dipole interaction. The paper is well organized and reasonably clearly written. I can recommend the publication of the manuscript in the journal Molecules, provided that the authors can address the following:

- The predicted structures of the molecules are sometimes bent or linear depending on the level of theory. What are the energy differences for the different structures from the highest level of theory? Is the highest level of theory expected to be sufficiently accurate in predicting the structures confidently? The authors shall also rule out the possibility that the obtained structures are not a local minimum found by the geometry optimizer.

I also have a couple of minor suggestions:

- The two-body channel and state crossing shall be mentioned in the introduction to provide the readers a more complete picture. And the discussion around line 47 is a bit difficult to follow: it is not immediate obvious hat are the significance of these numbers.

- The authors used color codes to show structural parameters obtained with different level of theory. I recommend adding legends to the figures so that readers could understand these more easily.

Author Response

  • The predicted structures of the molecules are sometimes bent or linear depending on the level of theory. What are the energy differences for the different structures from the highest level of theory? Is the highest level of theory expected to be sufficiently accurate in predicting the structures confidently? The authors shall also rule out the possibility that the obtained structures are not a local minimum found by the geometry optimizer.

Reply: As pointed out by the reviewer, the structures were predicted linear or bent depending on the level of theory. Using the highest-level of theory CCSD(T)/apqz, we have estimated that the energy differences between linear and bent structures are within 1 kcal/mol. This information has now been added to lines 87-89. As mentioned in the text, due to the small force constants along the Ng-O-B angle, the energy profile is quite flat. However, we did search other structures starting from very different geometries (nonplanar, smaller or larger bond angles, etc.) using MP2 and DFT methods. We are quite confident the reported structures are the lowest in energies.

  • - The two-body channel and state crossing shall be mentioned in the introduction to provide the readers a more complete picture. And the discussion around line 47 is a bit difficult to follow: it is not immediate obvious hat are the significance of these numbers.

Reply:We thank the reviewer for the suggestion. We have now mentioned the two-body channel and the state crossing at lines 56-59, and additional discussion (lines 49-51) has been made to the statement around line 47 in the original manuscript. 

  • The authors used color codes to show structural parameters obtained with different level of theory. I recommend adding legends to the figures so that readers could understand these more easily.

Reply: As suggested by the reviewer, the legends have been added to Figures 1-3.

Reviewer 2 Report

The manuscript by Tsai, Lu, and Hu, describes a theoretical study molecular anions containing rare gas atoms and small, highly electronegative molecules. It is an interesting and fairly well written text that identifies a number of potentially stable systems. Overall I think that it is a nice piece of work that would be well suited for publication in Molecules, but I do have a few comments and questions that I think should be addressed before it is accepted:

1) As a general remark, the text is well written. There are however numerous small errors in the English, mainly the result of missing words. For example: on line 22 there is an “in” missing immediately before the first occurrence of “the”; line 38 “… there is a rich …” etc. This occurs occasionally throughout the text and should be addressed. 

2) I might be missing something, but the S-T gaps shown in tables 1 and 2 do not match the numbers mentioned in the text. For example, on line 167 a gap of 15-23 kcal/mol is stated for OBOArO-, but in table 1 the corresponding values are around 40 kcal/mol. I’m also not sure how this value is defined when looking at figure 5. This should be made more clear.

3) As far as I can tell none of the energies reported include zero-point corrections. For such shallow energy minima this effect must be very important and could even in the lowest energy state be enough to dissociate the molecular ions (particularly for He). Do you have any estimates on the magnitudes of these corrections for the different systems relative to the binding energies that you report?

4) On a related note, are the calculations corrected for basis set superposition error (BSSE)? For such loosely bound systems this can give a significant contribution to the calculated energies. 

Author Response

1) As a general remark, the text is well written. There are however numerous small errors in the English, mainly the result of missing words. For example: on line 22 there is an “in” missing immediately before the first occurrence of “the”; line 38 “… there is a rich …” etc. This occurs occasionally throughout the text and should be addressed.

Reply: We thank the reviewer to pointed out the errors. We have done our best to correct these errors in the English.

2) I might be missing something, but the S-T gaps shown in tables 1 and 2 do not match the numbers mentioned in the text. For example, on line 167 a gap of 15-23 kcal/mol is stated for OBOArO-, but in table 1 the corresponding values are around 40 kcal/mol. I’m also not sure how this value is defined when looking at figure 5. This should be made more clear.

Reply: We are sorry that we did not make this clear. The crossing point for Ar is indeed only 6 kcal/mol higher than the singlet energy minimum as shown in Figure 5. The same crossing points have also been located using similar figures (data not shown) for Kr and Xe, and they are 15-23 kcal/mol higher than the corresponding singlet minima. The S-T gaps calculated at the singlet minima for Kr and Xe are 51-55 kcal/mol, which are ~10 kcal/mol higher than that of Ar.  We have now made additional discussion on lines  179-182.

3) As far as I can tell none of the energies reported include zero-point corrections. For such shallow energy minima this effect must be very important and could even in the lowest energy state be enough to dissociate the molecular ions (particularly for He). Do you have any estimates on the magnitudes of these corrections for the different systems relative to the binding energies that you report?

Reply: We thank the reviewer for the suggestion. We have now added the energies including zero-point energy corrections in Tables 1-2. The discussion for He has been added to lines 150-151.

4) On a related note, are the calculations corrected for basis set superposition error (BSSE)? For such loosely bound systems this can give a significant contribution to the calculated energies.

Reply: We agree that BSSE might make important contribution for MP2 using smaller basis sets. However, the way we calculate the three-body dissociation is relative to Y- + Ng + O, not relative to Y- + NgO. The effects are expected to be small. Furthermore, the discussion of relative energies is based on CCSD(T)/apqz which is not affected significantly by BSSE. This has been mentioned on lines 75-78.  

Reviewer 3 Report

The work "Theoretical Prediction on the New Types of Noble Gas Containing Anions OBONgO- and OCNNgO- (Ng = He, Ar, Kr and Xe)" by Cheng-Cheng Tsai et al. aimed at studying a peculiar class of non-covalent Noble gas (Ng) compounds by means of Post-SCF calculations aimed at clarifying their molecular structures and electron density distributions. As a whole the manuscript is of rather good quality with respect to the exposition of topics presented with a comprehesive approach to identify the theoretical aspect of the stability of this class of particular molecular systems. While the computational methods and tools used are at best for those kind of peculiar calculations the manuscript lacks some basic contribution regarding the bonds at play in such non-covalent compounds. In fact, while the approach to the stability studies of the various XNgO- cogeners results in a set of reliable molecular structure with respect to the literature, the usage of the electron density distribution to classify the bond character is quite primitive or at least, rather crude. Let me explain the concept by looking at a statement in "Conclusions": "The results show that they are similar to FNgO- with very short terminal NgO distances, and with the negative charge concentrated on the OBO (OCN, NCO) groups. The structures can conceptually be written as X-...Ng=O where the interaction between X and NgO is mostly ionic ...(tbc)" Assertion of this kind is only partially true and conceptually incorrect on such a kind of peculiar compounds where we have a large variety of non-covalent bonding motifs, ranging from purely (or nearly purely) van der Waals contacts to interactions featuring appreciable contributions of covalency and charge transfer (CT). I would suggest the authors to revise their study of the bonding characters of the molecular systems under study by following methods capable of identify the various contributions to chemical bonds at play before arriving at inaccurate conclusions. To this end, I strongly encourage the authors to have a look, among many others, at the paper: Borocci, S.; Grandinetti, F.; Nunzi, F. & Sanna, N. Classifying the chemical bonds involving the noble-gas atoms New Journal of Chemistry, Royal Society of Chemistry (RSC), 2020, 44, 14536-14550 and to carefully figure out how to proceed toward a comprehensive analysis of bond character of the molecular systems under study so to complete their work and made the manuscript of suitable quality for publication.

Author Response

I would suggest the authors to revise their study of the bonding characters of the molecular systems under study by following methods capable of identify the various contributions to chemical bonds at play before arriving at inaccurate conclusions. To this end, I strongly encourage the authors to have a look, among many others, at the paper: Borocci, S.; Grandinetti, F.; Nunzi, F. & Sanna, N. Classifying the chemical bonds involving the noble-gas atoms New Journal of Chemistry, Royal Society of Chemistry (RSC), 2020, 44, 14536-14550 and to carefully figure out how to proceed toward a comprehensive analysis of bond character of the molecular systems under study so to complete their work and made the manuscript of suitable quality for publication.

Reply: We thank the comments and suggestions by the reviewer. While it is certainly true there are more sophisticated bonding analysis methods, such as that recommended by the reviewer in the 2020 paper, the main focus of the manuscript is on the stability and the molecular structures of the noble-gas anions. And, frankly, with only 5 days to upload the revised manuscript, there is really not enough time to do additional extensive bonding analysis. We will certainly consider the suggested analysis in future works on noble-gas containing molecules. This has been explained on lines 202-206.

Reviewer 4 Report

This is another paper in a series of similar calculations, see e.g Refs 10-27. They are using accepted methodology and applying it to a novel system. I have no real qualms about this work, except I find it disappointing that they do not really provide anything obviously new to all the previous papers, and their conclusion is reminiscent of their 2005 paper.
What did puzzle me and I would like to see addressed is that they brought in some DFT calculations, without explaining why, especially the choice of MPW1B95 functional. It is not one I have seen used much and I am wondering why they chose it. More curious is that they point out a difference between the MP2 and DFT structures but do not discuss this anywhere. DFT is usually good at structure, if not energetics, and I would be interested in the authors' opinion on how the disagreement is arising.
Finally, when looking at such energy differences I would expect to see zero-point corrections, especially as they would be of the order of the differences being reported and they tend to have a significant effect for such weakly bound structures.

Author Response

What did puzzle me and I would like to see addressed is that they brought in some DFT calculations, without explaining why, especially the choice of MPW1B95 functional. It is not one I have seen used much and I am wondering why they chose it. More curious is that they point out a difference between the MP2 and DFT structures but do not discuss this anywhere. DFT is usually good at structure, if not energetics, and I would be interested in the authors' opinion on how the disagreement is arising.
Finally, when looking at such energy differences I would expect to see zero-point corrections, especially as they would be of the order of the differences being reported and they tend to have a significant effect for such weakly bound structures.

Reply: We humbly thank the reviewer for the comments and suggestions. In this work, we provided the energy and property criteria for the stability of YNgX- which, to our knowledge, have not been clearly discussed before. It was also not immediately clear before this study whether polyatomic anions Y- with delocalized charge can form stable noble-gas anions of the type Y-...Ng=O.

The selection of MPW1B95 was based on a recent benchmark study, and this has now been added on lines 68-69. The reasons for the different structures by MP2 and DFT and related issues has now been discussed on lines 86-89, 194-197. As suggested by the reviewer, the zero-point corrected energies have now been added to Tables 1 and 2, and discussed on lines 150-151 for Ng = He.